# Anti-Inflammatory Compounds from *Atractylodes macrocephala*

**DOI:** 10.3390/molecules24101859

**Published:** 2019-05-14

**Authors:** Dawoon Jeong, Guang-zhi Dong, Hwa Jin Lee, Jae-Ha Ryu

**Affiliations:** 1College of Pharmacy, Sookmyung Women’s University, Seoul 04310, Korea; water543@naver.com (D.J.); dgz77@hanmail.net (G.-z.D.); 2School of Industrial Bio-Pharmaceutical Science, Semyung University, Jecheon, Chungbuk 27136, Korea; hwalee@semyung.ac.kr

**Keywords:** nitric oxide, prostaglandin E_2_, inducible nitric oxide synthase, cyclooxygenase-2, nuclear factor-κB, *Atractylodes macrocephala*

## Abstract

In relation to anti-inflammatory agents from medicinal plants, we have isolated three compounds from *Atractylodes macrocephala*; **1**, 2-[(2*E*)-3,7-dimethyl-2,6-octadienyl]-6-methyl-2, 5-cyclohexadiene-1, 4-dione; **2**, 1-acetoxy-tetradeca-6*E*,12*E*-diene-8, 10-diyne-3-ol; **3**, 1,3-diacetoxy-tetradeca-6*E*, 12*E*-diene-8, 10-diyne. Compounds **1**–**3** showed concentration-dependent inhibitory effects on production of nitric oxide (NO) and prostaglandin E_2_ (PGE_2_) in lipopolysaccharide (LPS)-activated RAW 264.7 macrophages. Western blotting and RT-PCR analyses demonstrated that compounds **1**–**3** suppressed the protein and mRNA levels of inducible nitric oxide synthase (iNOS) and cyclooxygenase-2 (COX-2). Furthermore, compounds **1**–**3** inhibited transcriptional activity of nuclear factor-κB (NF-κB) and nuclear translocation of NF-κB in LPS-activated RAW 264.7 cells. The most active compound among them, compound **1**, could reduce the mRNA levels of pro-inflammatory cytokines (IL-1β, IL-6 and TNF-α) and suppress the phosphorylation of MAPK including p38, JNK, and ERK1/2. Taken together, these results suggest that compounds **1**–**3** from *A. macrocephala* can be therapeutic candidates to treat inflammatory diseases.

## 1. Introduction

*Atractylodes macrocephala*, is a perennial herb that has been used as a traditional medicine in Korea, China, and Japan for thousands of years to treat gastrointestinal dysfunction. Examples may include loss of appetite, abdominal distention, and diarrhea [1,2]. The *A. macrocephala* has been reported to possess diverse biological activities, including improved functional defects in digestive system, as well as anti-tumor, anti-inflammatory, anti-aging, anti-oxidative, and anti-bacterial activities [3,4,5,6,7]. *A. macrocephala* have been reported to contain sesquiterpenoids, polyacetylenes, phenylpropanoids, flavonoids, and polysaccharides [8]. Some sesquiterpenoids such as atractylenolide I~III exhibited inhibitory effect on LPS-induced NO production in macrophages and neuroprotective effects in the Parkinson’s disease model [8,9,10,11]. Moreover, polyacetylenes including atractylodemayne A, E, F, G, and 14-acetoxy-12-senecioyloxytetradeca-2*E*,8*E*,10*E*-trien-4,6-diyn-1-ol showed anti-inflammatory potential against LPS-induced NO production and carrageenan-induced paw edema without their detailed action mechanisms [8,12,13].

Inflammation is a complicated physiological response to various immune stimuli such as infection and tissue injury, which is regulated by diverse inflammatory mediators. Although highly regulated inflammatory response is beneficial for body functions, immoderate and deregulated inflammation can cause tissue damages or chronic diseases such as cancer, diabetes, obesity, and Parkinson’s diseases [14,15,16].

Macrophage, major immune effector cell, plays an essential role as a responder to inflammation. Macrophage can be stimulated by external pathogen such as lipopolysaccharide (LPS). The activated macrophage contributes and accelerates immune response to induce expression of pro-inflammatory cytokines and inflammatory mediators including nitric oxide (NO) and prostaglandin E_2_ (PGE_2_) [17]. NO is synthesized by nitric oxide synthases (NOS) such as neuronal NOS (nNOS), endothelial NOS (eNOS), and inducible NOS (iNOS). Large amounts of NO are mostly released by iNOS which can be highly expressed in chronic inflammatory disorders [18]. Alternatively, PGE_2_ is synthesized by cyclooxygenase 1 (COX-1) or cyclooxygenase 2 (COX-2). PGE_2_ is abundantly produced by upregulated COX-2 in inflammatory state, which leads to pain and inflammation [19]. Therefore, inhibition of inflammatory responses in activated macrophages can be a preventive strategy to diminish excessive inflammation.

During searching for anti-inflammatory principles from *A. macrocephala*, we isolated one quinone compound and two polyacetylenes, which have not been reported for anti-inflammatory activity. In these studies, we evaluated the anti-inflammatory effect of three active constituents from *A. macrocephala* in LPS-activated RAW 264.7 macrophage cells and investigated their underlying action mechanisms.

## 2. Results and Discussion

Nitric oxide (NO), a free radical molecule, plays an important role in physiological activities in vascular and neuronal functions in normal conditions. Under inflammatory responses, activated macrophages release high level of NO which causes deleterious and severe inflammation [20]. PGE_2_, a lipid mediator synthesized from arachidonic acid, is abundantly produced in inflammatory reaction. They potentiate inflammation through eliciting vasodilation and increase of local blood flow [21]. Therefore, reducing the NO and PGE_2_ levels in LPS-activated macrophage cells is reasonable approach to suppress inflammatory response.

We purified compounds **1**, **2**, and **3** from *A. macrocephala* as inflammatory modulators in LPS-activated macrophages by the activity-guided purification process. Their structures were elucidated by the analysis of mass and nuclear magnetic resonance (NMR) spectroscopic data [22,23]. Compound **1** has quinone moiety, while compounds **2** and **3** have acetylene moiety (Figure 1A).

Compounds **1**–**3** inhibited LPS-induced NO production in a concentration-dependent manner, whereas LPS treatment dramatically increased NO in RAW 264.7 macrophage cells (Figure 1B). The IC_50_ values of compounds **1**, **2**, and **3** were 3.7, 21.1, and 60.4 μM, respectively. We have also examined the effects of compounds **1**–**3** on LPS-induced PGE_2_ production. As shown in Figure 1C, compound **1** suppressed PGE_2_ production, while the other two compounds **2** and **3** showed marginal suppression (Figure 1C). Especially IC_50_ for PGE_2_ of the compound **1** was 5.26 μM. These findings guided us to examine the effect of compounds **1**–**3** on expression levels of iNOS and COX-2 which produce NO and PGE_2_ as key mediators of inflammation.

To examine whether the suppressed production of NO and PGE_2_ by compounds **1**–**3** were related with modulation iNOS and COX-2 expressions, western blot and RT-PCR analyses were carried out. As shown in Figure 2A, compounds **1**–**3** (10 μM) attenuated the LPS-induced iNOS protein levels in RAW 264.7 cells whereas LPS treatment showed enhanced protein level of iNOS. Compounds **1**–**3** also suppressed protein levels of COX-2 compared with LPS treatment. Moreover, compounds **1**–**3** down-regulated the iNOS and COX-2 mRNA levels in LPS-stimulated RAW 264.7 cells (Figure 2B). Compound **1** among three compounds showed the most potent inhibitory effect on iNOS and COX-2 expressions. The suppressive effects of compounds **1**–**3** on iNOS and COX-2 expressions were in parallel with the NO and PGE_2_ levels. These results suggest that compounds **1**–**3** suppressed the production of NO and PGE_2_ inflammatory mediators by inhibiting expressions of iNOS and COX-2 protein and mRNA in LPS-induced macrophage cell system. Thus, we hypothesized that a transcription factor for iNOS and COX-2 gene expression would be affected by compounds **1**–**3**.

To disclose the action mode of compounds **1**–**3** for inhibition of transcriptional expression of iNOS and COX-2, we examined the effect of compounds **1**–**3** on activation of nuclear factor κB (NF-κB) which controls the expression of iNOS and COX-2 [24]. First, we assessed the secretory alkaline phosphatase (SEAP) activity by the transcriptional activation of NF-κB in T-RAW 264.7 cells with pNF-κB-SEAP-NPT reporter gene systems. As seen in Figure 3A, compounds **1**–**3** significantly repressed the SEAP activity, while LPS treatment markedly increased the SEAP activity. This result indicates that compounds **1**–**3** can suppress transcriptional activity of NF-κB.

Next, we assessed the effects of compounds **1**–**3** on LPS-induced degradation of I-κBα. NF-κB is a dimeric transcription factor complexed with p50 and p65 subunits. In normal conditions, NF-κB exists in cytoplasm as an inactive form (p50/p65 dimer) by physically binding with inhibitor-κB (I-κB) [25]. However, I-κB is phosphorylated, ubiquitinated, and quickly degraded to discharge p50/p65 in inflammatory situations. Released p50/p65 moves to nucleus and initiates the expression of inflammatory genes such as iNOS and COX-2 [26]. Here, we observed the effects of compounds **1**–**3** on degradation of I-κBα and translocation of p65 subunit in LPS-stimulated macrophages. As shown in Figure 3B, LPS treatment showed very low level of I-κBα protein whereas compounds **1**–**3** recovered level of I-κBα protein. Furthermore, treatment of cells with compounds **1**–**3** decreased the level of p65 in nucleus. These results indicated that compounds **1**–**3** inhibited NF-κB activation by modulating degradation of I-κBα and nuclear translocation of p65. These findings proposed that anti-inflammatory potential of compounds **1**–**3** could be partially associated with their inhibition of NF-κB.

NF-κB has been known to regulate a large array of genes involved in inflammatory responses. In addition, NF-κB is required for an induction of large numbers of inflammatory genes, including those encoding IL-1, IL-6 and TNF-α as well as iNOS and COX-2 [27]. Thus, we investigated the effect of compound **1** on LPS-induced pro-inflammatory cytokines such as IL-1, IL-6 and TNF-α, because compound **1** is the strongest inhibitor of NF-κB among the three compounds. As shown in Figure 4, compound **1** concentration-dependently reduced mRNA expressions of IL-1, IL-6 and TNF-α in LPS-stimulated RAW 264.7 macrophage cells. These suppressive effects of compound **1** on pro-inflammatory cytokines came from inhibition of I-κB degradation and NF-κB nuclear translocation.

Mitogen-activated protein kinases (MAPKs) including p38, JNK, and ERK1/2 can regulate gene expression, cellular growth, and inflammatory response [28,29]. MAPKs can induce inflammatory cytokines in response to various inflammatory stimuli including LPS and TNF-α [30,31]. MAPKs signaling are positively associated with NF-κB activation [32,33]. To examine the effect of compound **1** on MAPKs signaling, we assessed the phosphorylation levels of MAPKs in LPS-activated macrophage cells. We observed the elevated phosphorylation of p38, JNK, and ERK1/2 by LPS treatment (Figure 5). However, compound **1** reduced the p38, JNK, and ERK1/2 phosphorylation in a concentration-dependent way with no alteration in total p38, JNK, and ERK1/2 levels. These results suggested that compound **1** could attenuate LPS-induced inflammation by suppression of MAPKs signaling pathway and NF-κB activation.

## 3. Materials and Methods

### 3.1. General Experimental Procedures

Mass spectra were obtained on a JEOL JMS-AX505WA mass spectrometer. NMR spectra were determined on a Varian UNITY INOVA 400 NMR spectrometer. Column chromatography was carried out over silica gel (40–60 μm, Merck, Kenilworth, NJ, USA). Fractions from column chromatography were monitored by thin layer chromatography (TLC) (silica gel 60 F_254_ and RP-C_18_ F_254S_, Merck) under UV light or by heating after spraying 10% H_2_SO_4_ in CH_3_OH (*v*/*v*).

### 3.2. Plant Materials, Extraction and Isolation

The dried rhizomes of *A. macrocephala* were purchased from the Kyungdong Herbal Market in Seoul, Korea. The air-dried material (10 kg) was extracted with 20 L of EtOAc three times. The EtOAc extract (310 g) was subjected to silica gel column chromatography eluting with n-hexane: EtOAc gradient system (100:1 to 1:2) to give 14 fractions. Fraction 9 (13.2 g), which suppressed NO production in LPS-stimulated RAW 264.7 cells, was further chromatographed on silica gel with n-hexane: acetone (30:1 to 1:2) to afford seven sub-fractions. By the activity-guided purification process, sub-fraction 9-6 (620 mg) was further purified by a silica gel column using n-hexane: EtOAc (30:1 to 1:2) to afford compound **1** (17 mg). Sub-fraction 9-4 (1.4 g) was re-chromatographed on silica gel with n-hexane: EtOAc (50:1 to 1:2) to yield compound **3** (53 mg). Fraction 12 (15.1 g) was chromatographed on silica gel with n-hexane: EtOAc gradient system (50:1 to 1:2) to afford 10 sub-fractions. Sub-fraction 12-6 (3.3 g) was further purified by a silica gel column with n-hexane: acetone (20:1 to 1:3) to yield compound **2** (45 mg). The structures of compounds **1**–**3** were identified as 2-[(2*E*)-3,7-dimethyl-2,6-octadienyl]-6-methyl-2,5-cyclohexadiene-1,4-dione (**1**), 1-acetoxy-tetradeca-6*E*,12*E*-diene-8,10-diyne-3-ol (**2**), and 1,3-diacetoxy-tetradeca-6*E*,12*E*-diene-8, 10-diyne (**3**) (Figure 1A) by spectroscopic analysis and comparison with the previously reported data [22,23]. Previously, compound **1** was reported from *A. lancea* [22] and *A. macrocephala* [34], and compounds **2** and **3** from *A. koreana* [23] and *A. chinensis* [35].

2-[(2*E*)-3,7-Dimethy-2,6-octadienyl]-6-methyl-2,5-cyclohexadiene-1,4-dione (**1**) yellow- brownish oil, HREIMS *m*/*z* 258.1646 (calculated for C_17_H_22_O_2_, 258.1620), ^1^H-NMR (CDCl_3_, 400 MHz): δ 6.55 (1H, dq, *J* = 2.7, 1.6 Hz, H-5), 6.47 (1H, dt, *J* = 2.7, 1.6 Hz, H-3), 5.15 (1H, t sext, *J* = 7.3, 1.2 Hz, H-2′), 5.08 (1H, t sept, *J* = 6.8, 1.2 Hz, H-6′), 3.13(2H, br d, *J* = 7.3 Hz, H-1′), 2.08 (2H, m, H-5′), 2.06 (5H, m, H-4′ and H-7), 1.70 (3H, br s, H-9′),1.62 (3H, br s, H-10′), 1.60 (3H, br s, H-8′); ^13^C-NMR (CDCl_3_, 100 MHz): δ 188.0 (C-1 and C-4), 148.5 (C-3), 145.9 (C-5), 140.0 (C-3′), 133.1 (C-6), 132.2 (C-2), 131.8 (C-7′), 123.9 (C-6′), 118.0 (C-2′), 39.6 (C-4′), 27.5 (C-1′), 26.4 (C-5′), 25.7 (C-9′), 17.7 (C-8′), 16.0 (C-10′), 16.0 (C-7).

1-Acetoxy-tetradeca-6*E*,12*E*-diene-8,10-diyne-3-ol (**2**) yellowish oil, HREIMS *m/z* 260.1448 (calculated for C_16_H_20_O_3_, 260.1413), ^1^H-NMR (CDCl_3_, 400 MHz): δ 6.33 (1H, dq, *J* = 15.0, 7.0 Hz, H-13), 6.27 (1H, dt, *J* = 15.0, 7.5 Hz, H-6), 5.58 (1H, dd, *J* = 15.0, 2.0 Hz, H-12), 5.55 (1H, dd, *J* = 15.0, 2.0 Hz, H-7), 4.37 (1H, m, H-1a), 4.11 (1H, m, H-1b), 3.7 (1H, m, H-3), 2.28 (2H, m, H-5), 2.07 (3H, s, Acetyl) 1.82 (3H, dd, *J* = 7.0, 2.0 Hz, H-14), 1.78 (1H, m, H-2a), 1.68 (1H, m, H-2b), 1.58 (2H, m, H-4); ^13^C-NMR (CDCl_3_, 100 MHz): δ 171.6 (aceetyl, C=O), 147.4 (C-6), 143.4 (C-13), 109.8 (C-12), 109.2 (C-7), 79.9 (C-11), 79.4 (C-8), 72.9 (C-10), 72.2 (C-9), 67.7 (C-3), 61.5 (C-1), 36.4 (C-2), 36.0 (C-4), 29.4 (C-5), 20.9 (acetyl, CH_3_), 18.9 (C-14).

1,3-Diacetoxy-tetradeca-6*E*,12*E*-diene-8,10-diyne (**3**) yellowish oil, HRFABMS *m/z* [M + H]^+^ 303.1468 (calculated for C_18_H_22_O_4_, 303.1596), ^1^H-NMR (CDCl_3_, 400 MHz): δ 6.33 (1H, dq, *J* = 15.0, 7.0 Hz, H-13), 6.25 (1H, dt, *J* = 15.0, 7.0 Hz, H-6), 5.57 (2H, br d, *J* = 15.0 Hz, H-7 and H-12), 5.00 (1H, tt, *J* = 7.0, 6.5 Hz, H-3), 4.10 (2H, t, *J* = 6.5 Hz, H-1), 2.09 (2H, br dt, *J* = 7.0, 7.0 Hz, H-5), 2.06 (6H, s, Acetyl), 1.89 (2H, dt, *J* = 6.5, 6.5 Hz, H-2), 1.84 (3H, dd, *J* = 7.0, 15.0 Hz, H-14) 1.70 (2H, m, H-4); ^13^C-NMR (CDCl_3_, 100 MHz): δ 170.9 (acetyl, C=O), 170.5 (acetyl, C=O), 146.5 (C-6), 143.5 (C-13), 109.8 (C-12), 109.5 (C-7), 80.0 (C-11), 79.2 (C-8), 73.1 (C-10), 72.3 (C-9), 70.3 (C-3), 60.6 (C-1), 33.0 (C-2), 33.0 (C-4), 29.1 (C-5), 21.0 (acetyl, CH_3_), 20.9 (acetyl, CH_3_), 18.9 (C-14).

### 3.3. Cell Culture

RAW 264.7 murine macrophage cells (ATCC, Rockville, MD, USA) were cultured in Dulbecco’s Modified Eagle Medium (DMEM) containing 10% FBS, 100 U/mL penicillin, and 100 μg/mL streptomycin (Life technologies, Frederick, MD, USA) at 37 °C in 5% CO_2_ in a humidified atmosphere.

### 3.4. Nitrite Assay

RAW 264.7 cells were stimulated with LPS (1 μg/mL) in the absence or presence of compounds for 20 h. NO released from cells was assessed by detecting nitrite in culture supernatant. Aliquots (100 μL) of culture media were incubated with 150 μL of Griess reagent (1% sulfanilamide, 0.1% naphthylethylene diamine in 2.5% phosphoric acid solution) at room temperature for 10 min. Absorbance at 540 nm was read by using a microplate reader (Molecular Devices, CA, USA). The concentration of NO was determined by the sodium nitrite standard curve.

### 3.5. PGE_2_ Assay

To observe the effects of compounds on COX-2, cells were seeded with aspirin (500 µM) to inactivate the COX-1. After 2 h, cells were washed with fresh media three times and incubated with LPS (1 µg/mL) in presence of compounds for 20 h. The PGE_2_ levels in culture media were determined using enzyme immunoassay kit (Cayman Chemical, Ann Arbor, MI) according to the manufacturer’s instruction. In brief, 50 μL of supernatant of the culture medium and 50 μL PGE_2_ tracer were put into the PGE_2_ EIA plate and incubated for 18 h at room temperature. The wells were washed with 10 mM phosphate buffer (pH 7.4) containing 0.05% Tween 20. Then 200 μL of Ellman’s reagent was added to the well and incubated in the dark. Following the developing step, the absorbance was read at 405 nm by a microplate reader. A standard curve was prepared simultaneously with PGE_2_ standard ranging from 0.06 to 6 pg/mL.

### 3.6. Western Blot Analysis

RAW 264.7 cells (5 × 10^5^ cells/60 mm dish) were treated with or without test compounds in presence of LPS (1 μg/mL) for 20 h. Cells were lysed gently with cell lysis buffer (Cell Signaling Technologies, Beverly, MA, USA). Cell lysates were centrifuged at 4 °C and supernatant were subjected to the quantitation of protein concentrations by the Bradford method. For preparation of cytosol and nuclear extracts, cells were treated with test compounds for 30 min prior to the stimulation with 1 μg/mL LPS. Following 15 min treatment of LPS, cells were collected by using NE-PER nuclear and cytoplasmic extraction reagents according to the manufacturer’s instructions (Pierce Biotechnology, Rockford, IL, USA). The protein lysates were then subjected to SDS-PAGE and transferred onto PVDF membranes. After blocking with 5% non-fat milk for 1 h, membranes were incubated with the primary antibody overnight at 4 °C. Antibodies against iNOS (BD Biosciences, Franklin Lakes, NJ, USA), COX-2 (Cayman Chemical Company, Ann Arbor, MI, USA), I-κB-α, p65 (Santa Cruz Biotechnology, Rockford, IL, USA), ERK1/2, phospho-ERK1/2, p38, phospho-p38, JNK, phospho-JNK, and β-actin (Cell Signaling Technology, Beverly, MA, USA) were used for immunoblot analysis. After incubation with the secondary antibody for 1 h at room temperature, proteins were detected by VersaDoc 3000 (Bio-Rad, Hercules, CA, USA) with ECL reagents (GE Health Care Life Sciences, Marborugh, MA, USA).

### 3.7. Reverse Transcription and Polymerase Chain Reaction (RT-PCR) Analysis

Cells (1.8 × 10^6^ cells/60mm dish) were stimulated for 6 h with LPS (1 μg/mL) in the presence or absence of test compounds. Total RNA was extracted by TRIzol (Life technologies, Frederick, MD, USA) according to the manufacturer’s instructions. RNA was reverse-transcribed into cDNA using reverse-transcriptase (Life technologies, Frederick, MD, USA) and random-hexamer (Cosmo, Seoul, Korea). The cDNA amplification was performed by using a recombinant Taq polymerase (Promega, Madison, WI, USA) and primers for iNOS, COX-2, IL-6, IL-1β, TNF-α, and β-actin. The amplified PCR products were separated on 2% agarose gels and stained with ethidium bromide.

### 3.8. Measurement of NF-κB Transcriptional Activity

NF-κB transcriptional activity was determined by using the stably transfected RAW 264.7 cells with pNF-κB-SEAP-NPT (T-RAW 264.7 cells) as described previously with some modifications [36,37]. T-RAW 264.7 cells were kindly gifted by Professor Yeong Shik Kim, Seoul National University, Korea. T-RAW 264.7 cells were plated on a 24 well plate overnight. Test compounds were added to cells 2 h before the treatment with LPS (1 µg/mL). After 16 h incubation, aliquots of culture medium were heated at 65 °C for 6 min and then the activity of SEAP (secretory alkaline phosphatase) was assessed. The transcriptional activity was expressed as relative fluorescence unit (RFU).

### 3.9. Statistical Analysis

All values were expressed as mean ± S.D. of three experiments. Statistical analysis was carried out with Student’s t-test. A *p*-value of <0.05 was considered as significantly different.

## 4. Conclusions

In the present study, we explored the anti-inflammatory potential of compounds **1**–**3** from *Atractylodes macrocephala*. Compounds **1**–**3** reduce NO and PGE2 production, and also suppress the iNOS and COX-2 expression in LPS-stimulated RAW 264.7 cells. The underlying mechanism proved that compounds **1**–**3** suppressed NF-κB through inhibiting I-κB degradation and the NF-κB nuclear accumulation. Moreover, compound **1**, the most potent among three components, decreased levels of pro-inflammatory cytokines including IL-6, IL-1β, and TNF-α, and also suppressed MAPKs phosphorylation in LPS-activated RAW 264.7 cells. Taken together, compounds **1**–**3** from *A. macrocephala* can be therapeutic candidates to treat inflammatory diseases.

## Figures and Tables

**Figure 1 molecules-24-01859-f001:**
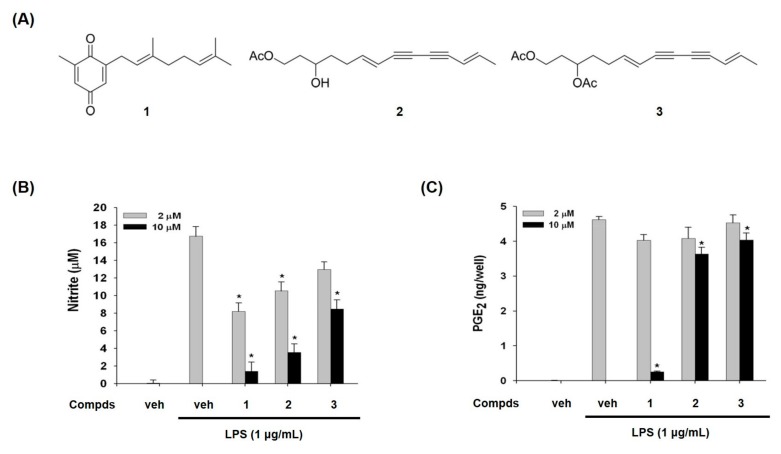
Chemical structures of compounds **1**–**3** from *A. macrocephala* (**A**) and effects of compounds **1**–**3** on lipopolysaccharide (LPS)-induced nitric oxide (NO) and prostaglandin E_2_ (PGE_2_)production (**B**,**C**). (B) NO released from cells was assessed as nitrite form in culture supernatant. The amount of nitrite in culture medium was measured by using the Griess reagents, as described in materials and methods. (C) The levels of PGE_2_ in culture medium were determined by enzyme immunoassay method. Veh means vehicle. The values are expressed as the mean ± S.D. of three individual experiments. * *p* < 0.01 indicate significant differences from the LPS alone.

**Figure 2 molecules-24-01859-f002:**
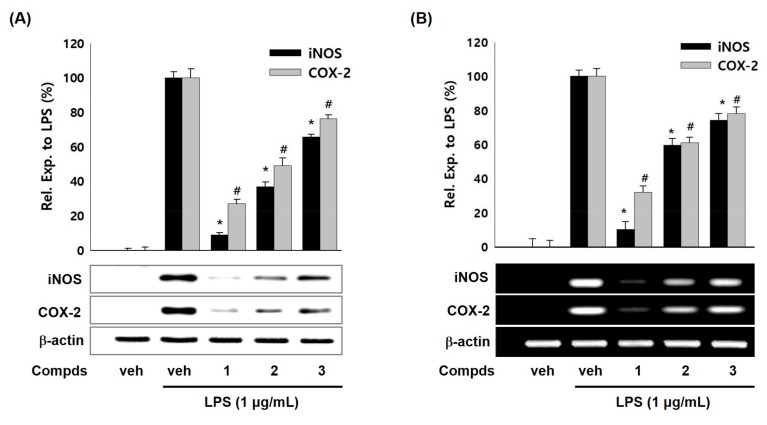
Effects of compounds **1**–**3** on the expression of LPS-induced iNOS/COX-2 protein and mRNA in RAW 264.7 macrophages. (**A**) Cells were activated with LPS (1 μg/mL) in presence or absence of compounds **1**–**3** (10 μM) for 20 h. Cell lysates were prepared and the iNOS, COX-2, and β-actin protein levels were determined by Western blotting. (**B**) Cells were treated with compounds **1**–**3** (10 μM) and/or LPS (1 μg/mL) for 6 h. The mRNA levels for iNOS, COX-2, and β-actin were determined by RT-PCR. The relative intensity of iNOS/COX-2 to β-actin bands was measured by densitometry. Veh means vehicle. The values represented mean ± S.D. of three individual experiments. *, ^#^
*p* < 0.01 indicate significant difference (* iNOS, ^#^ COX-2) from LPS alone.

**Figure 3 molecules-24-01859-f003:**
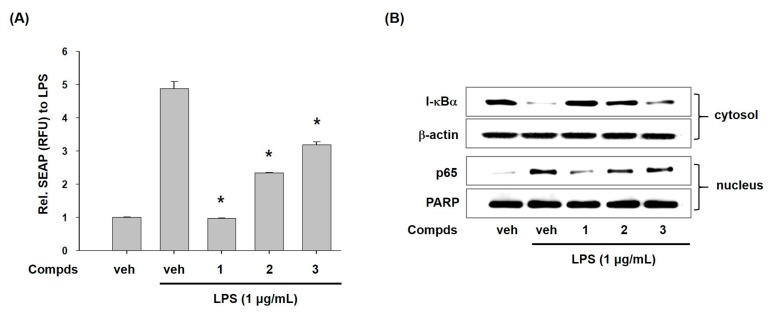
Effects of compounds **1**–**3** on LPS-induced NF-κB activation. (**A**) Effects of compounds **1**–**3** on LPS-induced nuclear factor-κB (NF-κB) transcriptional activation in T-RAW cells (the stably transfected RAW 264.7 cells with pNF-κB-SEAP-NPT. The pNF-κB-SEAP-NPT plasmid that permits expression of the secretory alkaline phosphatase (SEAP) reporter gene in response to the NF-κB activity and contains the neomycin phosphotransferase (NPT) gene for geneticin resistance.). T-RAW cells were treated with compounds 1–3 (10 μM) for 2 h prior to stimulation of LPS for 16 h. The transcriptional activity was assessed by measuring SEAP activity and then expressed as relative fluorescence unit (RFU). (**B**) Effects of compounds **1**–**3** on I-κB-α degradation and p65 translocation to nucleus in LPS-stimulated RAW 264.7 macrophages. Cells were pre-treated with compounds **1**–**3** (10 μM) for 30 min prior to LPS treatment for 20 min. Cytoplasmic and nuclear extracts were prepared for western blotting of I-κB-α and p65, respectively. β-Actin and poly ADP-ribose polymerase (PARP) were used as loading controls. Veh means vehicle. The values are presented as mean ± S.D. of three individual experiments. **p* < 0.01 indicate significant differences from the LPS alone.

**Figure 4 molecules-24-01859-f004:**
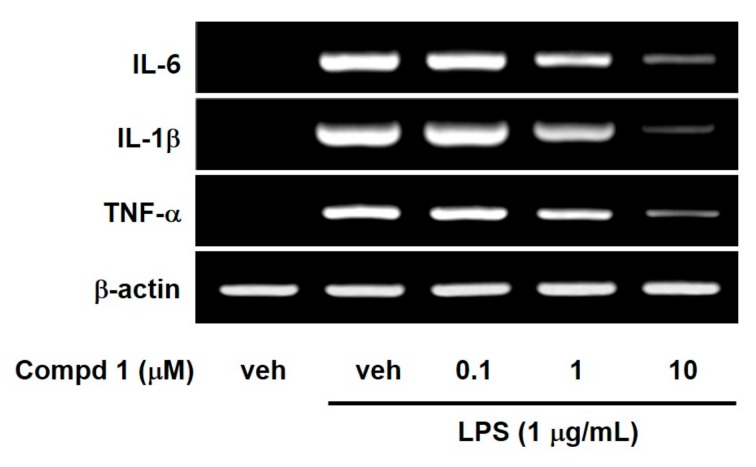
Effects of compound **1** on the expression of IL-6, IL-1β and TNF-α mRNA in LPS-activated macrophages. RAW 264.7 cells were treated for 6 h with compound **1** at indicated concentrations during LPS (1 μg/mL) activation. The mRNA levels of IL-6, IL-1β, and TNF-α were determined by RT-PCR. Veh means vehicle. Images are the representative results of three separate experiments.

**Figure 5 molecules-24-01859-f005:**
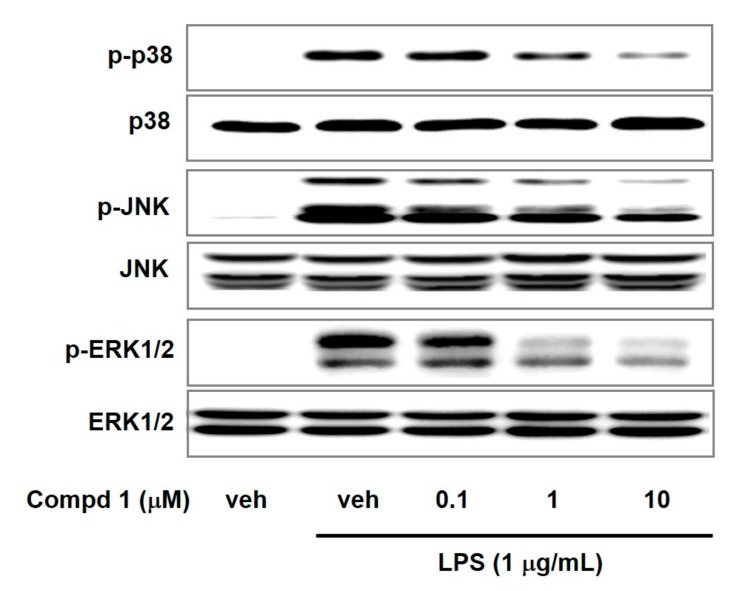
Effects of compound **1** on phosphorylation of p38, JNK, and ERK1/2 in LPS-activated macrophages. RAW 264.7 cells were pretreated with compound **1** (0.1, 1, and 10 μM) for 30 min and incubated further with LPS (1 μg/mL) for 15 min. The protein levels of ERK, JNK and p38 in cell lysate were determined by western blotting. Veh means vehicle. Images are representative results of three independent experiments.

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
