# Peer review of "Anti-Inflammatory Compounds from Atractylodes macrocephala"

_molecules, 2019, doi:10.3390/molecules24101859_

Reviewer 1 Report

The paper Anti-inflammatory compounds from Atractylodes macrocephala reports 3 anti-inflammatory compounds isolated from this medicinal plant ((2'E)-2-(3',7'-dimethylocta-2',6'-dienyl)-6-methyl-2,5-cyclohexadiene-1,4-dione; 1-acetoxy-tetradeca-6E,12E-diene-8,10-diyne-3-ol; 1,3-diacetoxy-tetradeca-6E,12E-diene-8, 10-diyne).  Those compounds exhibited concentration-dependent inhibitory effects on  the production  of  nitric  oxide  (NO)  and  prostaglandin E2 (PGE2)  in  LPS-activated  RAW  264.7  macrophages.  Western blotting and RT-PCR analyses demonstrated that the compounds suppressed the protein and mRNA levels of inducible nitric oxide synthase (iNOS) and cyclooxygenase-2 (COX-2).  Furthermore these compounds inhibited transcriptional  activity  of  nuclear factor-κB (NF-κB) and nuclear translocation of NF-κB in LPS-activated RAW 264.7 cells. Taken together, these results suggest that these compounds from A. macrocephala can be therapeutic candidates to treat inflammatory diseases. The manuscript is interesting and the research was well designed and performed, but there is need to point out what was done for the first time in present research (what is the novelty) in comparison to the existing papers (overview of the performed research on this plant with respect to phytochemistry and determined biological activities, in the context of chemical elucidation of the structures 1-3, their isolation protocol or regarding their determined biological activities). Before its acceptance, there is a need to answer to the remarks written further.

There is need to recheck the compound names and to take attention on E, Z symbols (isomers) to be written italic and there are extra spaces between the numbers in the compound names that should be corrected.

Page 1., lines 28-34: There is need to report more detail the identified compounds from Atractylodes macrocephala (not just by mentioning their general compound groups) and to present in short if there are data about structure-activity relationship of identified compounds. With this respect the following references are useful: Journal of Asian Natural Products research, 2017 Vol. 19, no. 1, 35–41; Journal of Ethnopharmacology 226 (2018) 143–167, others. There is need to significantly improve this part and to present major phytochemical present in this plant as well as their biological activities.

Page 1, lines 51-53: There is no data what was done for the first time in current research. Anti-inflammatory activity for 3 compounds were evaluated but there is need to report what was done for the first time and what is the overall novelty of performed research. In present way it is difficult to recognize the importance of the manuscript since the novelty is not clearly stated.

Page 6. Lines 170-185: Was the procedure of purification of compounds 1-3 done for the first time in present research or it was used from some other research.

Page 6., lines 185-208: Was the characterization of the compounds done for the first time or those compounds were previously characterized? If the compounds were first time characterized that there is need to add in the discussion part short discussion about NMR data and the corresponding structure. If the characterization was done before there is need to add adequate reference in the discussion part.

Author Response

We greatly appreciate the efforts of the Editorial Board and reviewers for giving us the valuable comments on our manuscript for Molecules. As you requested, we have enclosed a revised version of manuscript in response to the extensive and insightful reviewer’s comments. Included below is a point by point description of our responses to the reviewer’s comments. We hope that you and the reviewers find this revised manuscript acceptable for publication in Molecules. We are sending the revised manuscript. We did our best to satisfy the comments from reviewers.

 In revised manuscript, the corrected parts were highlighted in yellow and the deleted parts were colored in red and pointed as the deleted parts (a deletion line).

[Response to Reviewer’s comments]

· As reviewer mentioned, we corrected the compound name (compound 1) and wrote the E symbol in italic. Also, numberings of compound 1 were corrected. The corrected parts were highlighted in yellow. (Page 1, line 12-13; Page 7, line 198, line 203-209, line 211, line 219)

· Page1, line 28-34: As reviewer recommended, we improved this part with corresponding references and highlighted in yellow. (Page 1, line 34-39)

 · Page1, line 28-34: As reviewer commented, we revised this part to recognize the importance of this research and highlighted in yellow. (Page 2, line 56-58)

 · Page6, line 170-185: As reviewer commented, we revised and highlighted in yellow. (Page 7, line 201-202)

 · Page6, line 185-208: These compounds were previously reported in Atractylodes lancea, A. macrocephala, A. lancea, A. koreana. We described in page 7, line 201-202 with corresponding references and highlighted in yellow.

Reviewer 2 Report

The article is very interesting and well-written. However, I have a few questions and comments.

Page 2, lines 69-70 – „As shown in Fig. 1C, compounds 1-3 concentration-dependently suppressed PGE2 production (Fig. 1C)”. In my opinion, it should rather be written that compound 1 supressed production of PGE2, while the other two compounds 2 and 3 showed only little supression.

In the Figures, the description of the x-axis should be corrected, it is not known what the first column is.

On page 5 there are two Figures 4, while Figure 5 is missing.

On what basis the 9-6, 9-4 and 12-6 fractions were selected for further analysis.

Since the compounds presented are known, it should be clearly stated that they were isolated for the first time from this herb (if that's so).

Where is the description of the apparatus used for testing (Mass and NMR)?

I have great doubts about the description of NMR spectra. I am not sure that in the spectrum made with 400 MHz resolution you can see the long-range coupling very well. I would like to see the NMR spectrum of the compounds 1-3. In compound 1 protons H-6 and H-2 do not exist at all. There is no description of signals from protons H-3 and H-5. Is the signal from the CH3-9 group really a doublet?

Compounds 2 and 3 are very similar, in contrast to their NMR spectra. The description of the spectrum of compound 2 needs to be corrected.

Author Response

We greatly appreciate the efforts of the Editorial Board and reviewers for giving us the valuable comments on our manuscript for Molecules. As you requested, we have enclosed a revised version of manuscript in response to the extensive and insightful reviewer’s comments. Included below is a point by point description of our responses to the reviewer’s comments. We hope that you and the reviewers find this revised manuscript acceptable for publication in Molecules. We are sending the revised manuscript. We did our best to satisfy the comments from reviewers.

In revised manuscript, the corrected parts were highlighted in yellow and the deleted parts were colored in red and pointed as the deleted parts (a deletion line).

[Response to Reviewer’s comments]

 · Page 2, line 69-70: As reviewer mentioned, we revised as follows; compound 1 suppressed PGE2 production, while the other two compounds 2 and 3 showed marginal suppression (Fig. 1C). (Page 2, line 77-78).

 · Figures: As reviewer mentioned, we revised the x-axis of Figures (Fig. 1-5).

 · Page 5: As reviewer mentioned, we corrected. (Page 6, line 173, Figure 5)

 · As reviewer mentioned, we added activity-guided process in materials and method section as highlighted in yellow. (Page 6, line 189-192)

 · These compounds were previously reported in Atractylodes lancea, A. macrocephala, A. lancea, A. koreana. As reviewer mentioned, we revised and highlighted in yellow. (Page 7, line 201-202)

 · As reviewer mentioned, we added the general experimental procedures in in materials and method section as highlighted in yellow. (Page 6, line 179-184)

 · As reviewer mentioned, numberings of compound 1 were corrected. The corrected parts were highlighted in yellow (Page 7, line 203-209). We attached the NMR spectra of compound 1.

 · We observed one acetyl group in compound 2 (δ2.07, integral 30.21) and two acetyl groups in compound 3 (δ2.06, integral 62.26) from 1H-NMR spectra. The proton signal for H-3 of compounds 2 and 3 was observed at δ3.7 and δ 5.00, respectively.

From 13C-NMR spectra, we also observed carbonyl and methyl signals of each acetyl groups of compound 2 and 3 with proper chemical shift values. From all these data, we confirmed the chemical structures of compounds 2 and 3. We described the spectroscopic data and attached the NMR spectra of compounds 2 and 3.

Round  2

Reviewer 1 Report

The manuscript was improved.

Reviewer 2 Report

The authors answered all my questions. I do not have more comments.